# Catalytic Reduction of Dyes and Antibacterial Activity of AgNPs@Zn@Alginate Composite Aerogel Beads

**DOI:** 10.3390/polym14224829

**Published:** 2022-11-09

**Authors:** Fadila Benali, Bouhadjar Boukoussa, Nour-El-Houda Benkhedouda, Amina Cheddad, Ismail Issam, Jibran Iqbal, Mohammed Hachemaoui, Mohamed Abboud, Adel Mokhtar

**Affiliations:** 1Département de Génie des Matériaux, Faculté de Chimie, Université des Sciences et de la Technologie Mohamed Boudiaf, BP 1505, El-Mnaouer, Oran 31000, Algeria; 2Laboratoire de Chimie des Matériaux LCM, Université Oran 1 Ahmed Ben Bella, BP 1524, El-Mnaouer, Oran 31000, Algeria; 3Department of Chemical Engineering, Khalifa University of Science and Technology, Abu Dhabi P.O. Box 127788, United Arab Emirates; 4College of Natural and Health Sciences, Zayed University, Abu Dhabi P.O. Box 144534, United Arab Emirates; 5Département de Sciences de la Matière, Institut des Sciences et Technologies, Université Ahmed Zabana, Relizane 48000, Algeria; 6Catalysis Research Group (CRG), Department of Chemistry, College of Science, King Khalid University, P.O. Box 9004, Abha 61413, Saudi Arabia; 7Département Génie des Procédés, Institut des Sciences et Technologies, Université Ahmed Zabana, Relizane 48000, Algeria

**Keywords:** alginate, AgNPs, nano-catalyst, dye reduction, antibacterial activity

## Abstract

This work focuses on the preparation of aerogel composite beads based on Zn(II)-crosslinked alginate and loaded with different percentages of AgNPs using a simple approach. The obtained samples were evaluated in two different applications: the first application consists in their use as catalysts for the reduction of MB, MO, OG and CR dyes in a simple and binary system under the presence of NaBH_4_. For this, several parameters affecting the catalytic behavior of these catalysts have been investigated and discussed such as the catalyst mass, AgNPs content, dye nature, and the selectivity of the catalyst in a binary system. The second application concerns their antibacterial activities towards two Gram-negative bacteria *Escherichia coli* (ATCC 25922), and *Pseudomonas aeruginosa* (ATCC 27853), and a Gram-positive bacteria *Staphylococcus aureus* (ATCC 25923). The physico-chemical properties of different samples were characterized by XRD, FTIR, SEM/EDS, and TGA analysis. The obtained results confirmed the presence of AgNPs on a highly porous alginate structure. The dispersion of a high percentage of AgNPs leads to the formation of nanoparticles on the outer surface of the alginate which led to their leaching after the catalytic test, while the composite having a low percentage of AgNPs showed good results through all dyes without leaching of AgNPs. For the antibacterial application of the different samples, it was shown that a composite with a higher percentage of AgNPs was the most effective against all bacteria.

## 1. Introduction

Improving water quality is one of the major concerns of all living species. In recent years, a significant increase in pollution has been observed in the industry field, particularly in the textiles industry, which generated significant discharges, most of them being dyes or toxic chemicals [1,2]. It is in this context many studies have been devoted to the development of simple, ecological, and effective methods for the elimination of these toxic pollutants [2,3,4,5,6]. Among the most used methods for the remediation of contaminated water are photocatalysis, oxidation, reduction, membrane filtration, adsorption, ozonation, coagulation, flocculation, and other processes. 

Recently the reduction of organic pollutants by nanoparticles has shown many advantages where the reaction process can be achieved in a shorter reaction time, thus the obtained products can be used in other chemical processes. Silver nanoparticles have shown their efficiency for this type of reaction, their role lies in the transport of electrons from the donor (reducing agent) to the acceptor (pollutant). However, silver nanoparticles suffer from an aggregation problem due to the interactions that exist between them, which reduces their use in the catalysis field. To overcome this problem, several supports have been used such as zeolites [7,8], mesoporous silica [9], polymers [10], activated carbon [11,12], clays [13], metal oxides [14], and MOFs [15] to stabilize the metal nanoparticles.

Polymers have been considered essential materials in our lives due to their wide range of applications such in food, automotive, packaging, pharmaceuticals, agriculture, aeronautics, cosmetics, biomedical, optoelectronics, catalytic supports and also as useful materials for the elimination of pollutants due to their functional groups which are considered as active sites [16,17,18,19]. Very particular attention has been targeted towards biopolymers due to their compatibility with living systems by constituting a good alternative to polymers [20,21,22]. Polysaccharides are sustainable, environmentally friendly and ubiquitous biological materials from which they are found in algae (e.g., alginate), animals (e.g., chitosan), plants (e.g., starch, cellulose, pectin, guar gum) and microbes (e.g., dextran); these biomaterials comprise a monosaccharide repeating unit and a large number of functional groups. Among the remarkable properties of these compounds is that they present a gelling potential, strong hydrophilic character and rigid structure, and large surface area [20]. Thus, recent work on this type of material has led to new applications in the biomolecular and nanomaterial fields [23,24,25,26,27,28,29,30].

In the last years, polysaccharides have been considered as ideal carriers for the stabilization of metallic nanoparticles MNPs [31,32,33,34,35]. The presence of functional groups can lead to a good dispersion of nanoparticles on their surface, which makes them efficient catalytic supports for a wide variety of reactions [31,32,33,34,35]. These biomaterials can easily form gels in the presence of divalent or trivalent cations, by ionic crosslinking and coordination between functional groups and metal ions, which can broaden their application field, particularly in the catalysis and treatment of bacteria [8,33,36,37]. As they have the ability to protect metallic nanoparticles against aggregation [8,31,35,38,39,40]. Among the polysaccharides, alginate has become one of the major interests of scientific research, particularly in the field of nanomaterials. Thus it is easier to control their shapes (beads, films, etc.) which facilitates their recovery compared to other supports having ultrafine particles which require other treatments for their recovery during the reuse process [22,41]. This kind of material can easily activate nucleophilic and electrophilic reactions by carboxylate groups and hydrogen bonding as a bifunctional heterogeneous organocatalyst [42,43,44]. Alginate hydrogels have different physicochemical properties which strongly depend on the nature of the crosslinking ion, the source of alginate, the concentration, and the gelling method used. Following these properties mentioned above, alginate-based materials have been used in several fields such as biomedicine, biology, catalysis, adsorption, and separation [17,22,45,46,47]. The most important property of alginates is their ability to react with divalent and trivalent cations to form beads and films of hydrogels, resulting in a stable and ordered three-dimensional network described as the “egg box” model [42,48,49]. Their properties differ depending on the nature of the crosslinking agent used, which can significantly influence their applications, particularly in catalysis and in the treatment of bacteria.

This paper deals with the dispersion of silver nanoparticles on alginate crosslinked by a zinc divalent cation. Zinc was chosen as the cross-linking agent because of its antibacterial properties and its diverse involvement in innate and adaptive immune response processes [50,51,52]. Zinc also exhibits strong catalytic activity in electrochemistry, oxidation process and is considered as a biomaterial in biodegradable implantation. AgNPs are considered as antibacterial agents, antiseptics and excellent reducing agents in nanocatalysis [9,53,54]. Therefore, the use of a bimetallic composite can improve the performance of the resulting material towards antibacterial applications due to the effect of the synergy between Zn^2+^ and AgNPs species which have action towards different bacterial strains.

The objective of this work is to synthesize nanocomposite materials based on alginate cross-linked by Zn^2+^ and supported by AgNPs silver nanoparticles with different percentages using a simpler and greener approach. These composite beads have been explored as an environmentally friendly heterogeneous catalyst for the reduction of dyes in a simple and binary system using NaBH_4_ as a reducing agent. To understand the effect of the Zn^2+^ crosslinking agent and AgNPs, another application was used which consists of the application of the composite beads in the antibacterial activity towards three types of bacteria. All the obtained results were correlated according to the content of AgNPs, Zn^2+^ and their activities.

## 2. Experimental

### 2.1. Chemicals and Reagents 

Sodium Alginate (low density, Sigma Aldrich, Saint Louis, MO, USA), Zinc Nitrate (Zn(NO_3_)_2_, Riedel-deHaen, Seelze, Germany), Silver Nitrate (AgNO_3_, Riedel-deHaen), Sodium Borohydride (NaBH_4_, 98%, Sigma-Aldrich), Methylene Blue (MB, C_16_H_18_ClN_3_S, Genaxis Biothechnology, Vadodara, India), Methyl Orange (MO, C_14_H_14_N_3_NaO_3_S), Orange G (OG, C_16_H_10_N_2_Na_2_O_7_S_2_), and Congo Red (CR, C_32_H_22_N_6_Na_2_O_6_S_2_) were supplied from Sigma-Aldrich. Distilled water was involved throughout all the experiments.

### 2.2. Preparation of Zn–Alginate(AgNPs)

The synthesis of the nanocomposite beads Zn–AlG(AgNPs) was carried out in the following steps: firstly, a 2% alginate solution was prepared in 100 mL of distilled water and agitated vigorously. After homogenization of the solution the mixture was loaded into a sterile syringe and dripped under magnetic agitation into a zinc solution (2 g of Zn(NO_3_)_2_ dissolved in 100 mL of distilled water). The latter plays the role of a crosslinking agent, each drop is transformed into transparent bead by complete crosslinking. These beads remained in a zinc solution for 24 h. Finally, the hydrogel beads were rinsed with distilled water to remove excess of Zn^2+^. Afterward, these beads were added in a solution of silver nitrate AgNO_3_ with different concentrations (0.5%, 1%, 2%, and 3%) for two days, then were washed three times with distilled water. These hydrogel beads were dried by lyophilization for 12 h and transformed into aerogel beads.

All aerogel beads loaded with Ag^+^ were treated with freshly prepared solution of NaBH_4_ (1M). A black color was observed once these beads were added to the NaBH_4_ solution, which confirms the reduction of Ag^+^ ions to (Ag^0^NPs). The obtained materials were named as follows: Zn–AlG(Ag 0.5%), Zn–AlG(Ag 1%), Zn–AlG(Ag 2%), Zn–AlG(Ag 3%) according to the percentage of silver used, and a Zn–AlG without silver was kept as a reference sample. The preparation steps are well detailed in Figure 1. 

### 2.3. Catalytic Test

The catalytic reduction of the designed catalyst was studied towards different dyes under the presence of NaBH_4_ as model reactions. The procedure is very similar to previously published work, but with slight modifications [36,55,56]. For this, two systems have been investigated based on the simple and binary systems, to determine the selectivity of obtained materials. For the simple system a concentration of 0.1 mM of dye was used in each test. Firstly 1.5 mL of dye solution and a catalyst amount was added in a quartz cuvette and then 1.5 mL of freshly prepared NaBH_4_ (8 mM) was added. The cuvette containing the reaction mixture was placed in a UV-vis (Specord 210 Analytik Jena UV–vis instrument) in which the reaction was monitored in situ and every 30 s an analysis was carried out. Several parameters affecting the reaction were optimized and discussed such as the effect of the AgNPs content, the catalyst mass, and the dye nature (MB, CR, MO, and OG).

For the binary system, three systems are selected as model reactions (MB+OG, MB+MO, and MB+CR); each system contains the MB dye and one of the other azo dyes (CR, MO, and OG). For this, the best catalyst optimized under the previous conditions was used in this study, using the following operating conditions: [dyes] = 0.05 mM; V_dyes_ = 1.5 mL; [NaBH_4_] = 8mM; V_NaBH4_ = 1.5 mL, and the catalyst mass = 5.3 mg. 

The dye conversion was calculated using Equation (1) and rate constant k_app_ (s^−1^ or min^−1^) was determined graphically using first-order kinetics as shown in Equation (2). Where *C*_0_ and *C_t_* represent the initial and final dye concentration, respectively.
(1)$$Conversion\;\left(\%\right)=\frac{C_{0}-C_{t}}{C_{0}}\times 100$$
(2)$$Ln\left(\frac{C_{t}}{C_{0}}\right)=-k_{app}\times t$$

### 2.4. Evaluation of Antibacterial Activity

The antibacterial action of Zn–AlG(Ag NPs) nanocomposite was evaluated by the conventional disc method [57]. For this study, some bacterial pathogens were used. Gram-negative and gram-positive bacteria are as follows: *Escherichia coli* (ATCC 25922), *Pseudomonas aeruginosa* (ATCC 27853), and Gram-positive bacteria *Staphylococcus aureus* (ATCC 25923) were cultured on Mueller–Hinton agar at 37 °C for 18 h. The composites were modeled in the form of discs using a Perkin-Elmer pelletizer (Waltham, MA, USA). The first step entails preparing the inoculum from a pure culture of the bacteria that will be tested on the isolation medium, scraping a few perfectly isolated colonies with a sterile swab, well-homogenizing the bacterial suspension, and then adding them to 9 mL of sterile physiological water at 0.9%, its opacity must be equivalent to 0.5 McFarland (10^6^ CFU). Once the Muller–Hinton agars are inoculated with a pure culture of the strain to be studied in the kneaded dishes, the discs of each composite at different concentrations are placed on the surface of the agar using sterilized forceps, and incubated at 37 °C for 24 h. The effects of the compounds are measured by the size of the distinct halo of inhibition that surrounds the contact zone. The antibiotic gentamicin and the parent material Zn–AlG were also used to compare them with the obtained materials.

### 2.5. Characterization of Composite Beads

All samples were analyzed by Fourier transform infrared spectroscopy (FTIR) using Bruker alpha Platinum-ATR instrument. The designed materials were characterized by XRD analysis using a Bruker D8 powder diffractometer (Cu-Kα radiation). A thermobalance (Perkin-Elmer STA 6000) was used to measure the thermal stability and mass losses of obtained samples under nitrogen flow. Scanning electron microscopy coupled with energy dispersive X-ray spectroscopy (SEM-EDS) was used to evaluate the morphology of obtained samples using a TESCAN VEGA (LMU) SEM with an INCAx-act (Oxford Instruments, Concord, MA, USA). 

## 3. Results and Discussion

### 3.1. Characterization of Materials

#### 3.1.1. XRD

The XRD patterns of parent material Zn–ALG and Zn–ALG(AgNPs) containing different percentages of AgNPs are shown in Figure 2. The parent material presented some weak peaks between 2θ = 13, 45°, and 22.50° mainly due to the strong interactions generated between its chains by intermolecular hydrogen bonding. These peaks correspond to the reflections of the following planes (110) and (200) located at 2θ = 13.45° and 22.50°, respectively [58]. The modified materials show new peaks characteristic of AgNPs; the peaks observed at 2θ = 38.34°, 44.18°, 65.55°, and 77.17° correspond to the following reflections (111), (200), (220), and (311), respectively, which confirms the good formation of AgNPs with a face-centered cubic structure [59]. It is observed that the increase in the AgNPs content on the surface of the Zn–AlG leads to the formation of a crystalline composite, which is mainly due to the covering of the external surface of the alginate by AgNPs.

#### 3.1.2. FTIR

The FTIR spectra of different materials are shown in Figure 3. All spectra are identical compared to the parent material (Zn–AlG) but with slight modifications. Alginate is characterized by a broad band at 3269 cm^−1^ which is attributed to the stretching of the –OH groups present in the polymer chain as well as in the physisorbed water molecules [8,36,40], the weak band around 2922–2895 cm^−1^ is attributed to the vibration of –CH bonds. The low bands around 1567−1591 cm^−1^ and 1405−1415 cm^−1^ are attributed to the stretching of COO^−^, these bands overlap with characteristic bands of –OH [8,36,40]. It is clear that these bands shifted slightly when increasing the AgNPs content, probably due to several interactions between the silver nanoparticles and the functional groups of the alginate [55]. Interactions between alginate and silver (before NaBH4 treatment) can form different metallic complexes. According to Nakamoto, the most probable structures are uncoordinated or ionic structures, bidentate chelating structures, unidentate structures, and bidentate bridging structures [60].

#### 3.1.3. TGA

The samples were analyzed by thermogravimetric analysis in order to study the thermal behavior of obtained samples. As shown in Figure 4, all materials exhibited three stages of degradation, the first stage of degradation takes place in the range of 30 to 120 °C which is mainly due to dehydration of physisorbed water molecules [8,36,40]. These results are in agreement with those obtained by the FTIR which confirmed the presence of physisorbed water molecules. This hydrophilic character is the result of several interactions between the water molecules and the functional groups containing composite beads (−COO^−^, and −OH). 

The second stage is characterized by the degradation of the alginate biopolymer (between 120−400 °C) which leads to the formation of intermediate products. In this step, about 37% of mass losses were obtained for the samples Zn–AlG(Ag 0.5% and 3%), and about 7% for Zn–AlG(Ag 1% and 2%). At the third stage of degradation (>400 °C), about 16% mass loss was obtained for all the composite Zn–AlG samples (Ag 0.5% and 3%), and 22% for Zn–AlG(parent material), and Zn–AlG(Ag 1% and 2%) which is mainly due to the degradation of the intermediate products of the second stage [8,36,40].

#### 3.1.4. SEM

Figure 5 presents the images of the internal and external morphology of the composite aerogel beads at different magnifications. The dimensions of the aerogel beads remained the same after the lyophilization treatment, the external surface of the beads became rough due to the dehydration of water molecules after the lyophilization treatment [8,36,40]. This shows that many pores appeared making the beads poly-porous structure, this also confirms the cross-linking of alginate by Zn^2+^. It is clear that the parent material and its modified counterparts presented a homogeneous surface and no trace of AgNPs was observed mainly due to their good dispersion (results in agreement with the EDS mapping analysis) in the alginate matrix except for the case of composite beads Zn–AlG(Ag 3%) which has presented AgNPs on its external surface.

#### 3.1.5. EDS

The EDS characterization made it possible to confirm the presence of Zn^2+^ ions in the alginate matrix as well as the silver after treatment of the beads. EDS analysis (Figure 6, (Zn–AlG)) showed that the beads were mainly composed of oxygen, carbon and 4.71% zinc. The composition of beads is very similar from the point of view of their chemical structure. After modifying the parent material Zn–alginate by different percentages of AgNPs, it was found that the content of the latter increases with the increase of AgNPs. These results are in agreement with those obtained by XRD analysis.

### 3.2. Reduction of Dyes

#### 3.2.1. Reduction of Organic Pollutants in a Simple System


(a)Effect of the catalyst mass


The mass of catalyst represent a very important role in the conversion of reactants [15,55,61,62,63]. For this reason, a study was carried out on the reduction of MB dye as a prototype reaction using the following conditions: Zn–AlG(Ag 3%) catalyst, [MB] = 0.1 mM, [NaBH_4_] = 8 mM, the mass of catalyst was varied between 2–5.3 mg. The obtained results are presented in Figure 7. 

Figure 7a shows the repetitive UV-vis spectra of the MB dye during the reduction reaction. This material (Zn–AlG(Ag 3%)) exhibits significant catalytic activity towards the MB dye in which the reaction time does not exceed 840 s. It should be noted that this reaction took place at a lower concentration of NaBH_4_ showing its performance. However, during the addition of catalysts based on metallic nanoparticles, a strong decrease in the intensity of the MB dye was observed due to the hydrogenation process. The reduction of the MB dye by Zn–AlG(Ag 3%) essentially leads to the formation of leuco-MB which is colorless and less toxic compared to the MB dye. This product is characterized by the band located at 257 nm. The disappearance of the bands at 292 nm and 613 nm and 664 nm confirms the total conversion of MB dye into Leuco-MB [8,40,63]. 

Figure 7b shows the correlation between catalyst mass, reaction time, and MB dye conversion. It is clear that the masse of Zn–AlG(Ag 3%) significantly influences the reaction time also the conversion of MB dye. Better conversion was obtained only in 240 s when using 5.3 mg of Zn–AlG(Ag 3%). This behavior is related to the increase in the number of sites compared to the number of molecules of MB when increasing the mass of catalyst, which implies a rapid reduction. These results are in agreement with the literature [15,61,62]. 

Comparison of these results with other materials conducted in the literature clearly shows the performance of this system to reduce MB dye in shorter reaction time and also at low NaBH_4_ concentration. It must also be taken into account that the rate constant was greater than the values indicated in the Appendix A, which confirms a good reduction of MB dye has been obtained only at a low concentration of NaBH_4_ and low mass of catalyst [31,34].
(b)Effect of silver content

In this part, effect of the AgNPs content was carried out by varying only the catalyst (Zn–AlG(Ag 3%), Zn–AlG(Ag 2%), Zn–AlG(Ag 1%), Zn–AlG(Ag 0.5%) and Zn–AlG and keeping the other parameters constant. As shown in Figure 8, it is clear that the nature of the catalyst significantly influences the conversion of MB dye and the reaction time. According to this figure, the presence of AgNPs in the catalyst accelerates the process of the reduction in which the catalyst Zn–AlG(Ag 3%) presented the best performance. However, it was found that strong leaching of AgNPs takes place when using catalysts with higher content of AgNPs. So in terms of stability, the Zn–AlG(Ag 0.5%) catalyst was the most stable, suggesting that low AgNPs contents are chemically bonded with the biopolymer (due to the presence of strong sites), but at higher levels of AgNPs it is assumed that they are just deposited on the surface by weak interactions (Hydrogen, and Vander walls). Therefore, the Zn–AlG(Ag 0.5%) catalyst was used for the rest of our work due to its stability.
(c)Effect of the nature of the dyes

The selectivity of a catalyst differs from one pollutant to another, this is why we have used different azo dyes (OG, MO and CR) while maintaining the conditions previously optimized, the obtained results are represented in the Figure 9. According to these Figure 9a–d, it is clear that the catalyst Zn–AlG(Ag 0.5%) represented better results in terms of dyes conversion. However, it was more efficient with the OG dye in which its degradation was faster compared to the other dyes. A reaction time of 480 s was sufficient for the total degradation of the OG dye. 

The comparison of the rate constant of this catalyst has shown good results in terms of efficiency [31,33,34,36,37], using only lower concentration of reducing agent NaBH_4_ (see Appendix A).

#### 3.2.2. Reduction of Organic Pollutants in a Binary System

Reduction in a binary system is one of the least studied reactions, this reaction model can give more information about catalyst selectivity. To study this reaction, three reaction models were studied based on a mixture between the MB dye and another azo dye (MO, OG, and CR). As shown in this Figure 10, our catalytic system showed significant selectivity and total conversion towards the cationic MB dye. Thus, for the other azo dyes, a low conversion was observed. This behavior is linked to the nature of the surface of the catalyst [7,64]. It is necessary to take into account that the addition of NaBH_4_ leads to the formation of an electronic layer on the surface of the nanoparticles which generates attraction forces between the nanoparticles and the cationic MB dye subsequently leading to a faster reduction of the latter. These results are in agreement with the literature [7,64].

#### 3.2.3. Catalyst Reuse

The Zn–AlG(0.5% Ag) catalyst was used in six consecutive cycles toward the reduction of MB dye as a model reaction (see Figure 11). In each reuse, the catalyst was recovered easily (due to its shape and size), washed only with water, and then tested for another cycle. Figure 11 clearly shows that this catalyst was effective in the reduction of the MB dye in which the conversion of MB dye was almost total in each reuse. However, it was necessary to increase the reaction time up to 30 min to reduce the total MB during the sixth reuse. Similar behavior has been observed in previously published work [14].

#### 3.2.4. Mechanism of Reduction

The mechanism of reduction of OG, MB, CR, and MO has been investigated in several studies (see Figure 12), in general, the first step consists of the diffusion of reactants inside the pores of the catalyst, as it has been shown in this study, the first step was characterized by a large induction zone due to the tortuous diffusion of reagents inside the pores (see Appendix A). After dissociation and diffusion of NaBH_4_ electronic layers and hydride species form on the surface of silver nanoparticles which causes interactions with organic pollutants. The second step concerns the transport of electrons from NaBH_4_ to the organic pollutant acceptor. At this step, several intermediate hydrogenation reactions can take place caused by the transfer of electrons and the active H species (coming from the cleavage of the H-B) leading subsequently to the formation of the desired products. The reduction of azo dyes under the action of a catalyst and the reducing agent NaBH_4_ leads in the first step to the formation of an unstable intermediate product (hydrazine) which will be subjected to a second hydrogenation subsequently leading to the formation of amino derivatives [65,66]. As shown in Figure 9, the presence of new bands around 249 nm confirms the presence of amino derivatives [63]. For the reduction reaction of the dye MB to Leuco-MB, the reaction takes place in the groups –N=C– and -C=N^+^(CH_3_)_2_ which leads to the formation of –NH–CH– and –CH–HN^+^(CH_3_)_2_, respectively [67,68]. The hydrogenation of the MB dye generates the formation of a new band located around 258 nm in the UV-vis spectrum characteristic of the presence of Leuco-MB product.

### 3.3. Antibacterial Properties

According to the results mentioned below, it turned out that a zone of inhibition around the discs was observed against *Escherichia coli*. This bacterium does not have a very high resistance potential against all the materials presented in Table 1 and Appendix A. A salient fact could emerge from these results, and which concerns the Pseudomonas aeruginosa strain. This bacterium showed a certain variable sensitivity for these synthesized materials. However, it was found that the parent material Zn–AlG showed no activity via this bacterium. The Zn–AlG(Ag 1%) sample presented moderate activity with an inhibition diameter of 16 mm. It should be noted that the highest antibacterial activity was recorded by Zn–AlG(Ag 2%), with an inhibition halo value of 22 mm. So for the materials, Zn–AlG(Ag 0.5%), Zn–AlG(Ag 3%), and (Na–AlG) showed a inhibition zone between 20–19 mm. These results indicated that *Escherichia coli* was more sensitive than Pseudomonas aeruginosa germs. Almost the same remarks were observed for the S.a bacterium but the materials Na–AlG, Zn–AlG(Ag 3%), and Zn–AlG(Ag 1%) were the most efficient, hence an inhibition zone of approximately 19–18 mm was obtained but it is still lower compared to the antibiotic Gentamycin.

This antibacterial activity is linked to the synergistic effect between Zn^2+^ and AgNPs and as well as the functional groups containing alginate [9,69,70,71]. According to the literature, Zn^2+^ and AgNPs have been considered as good antibacterial agents for a variety of bacteria. The activity of different materials lies in the ease of diffusion of Zn^2+^ and Ag^+^ metal ions (due to the partial dissolution of AgNPs) which are considered to be the major component linked to the antibacterial action [72].

## 4. Conclusions

The Zn–AlG composite beads was well prepared by crosslinking the alginate with Zn^2+^ ions. The hydrogel beads were subjected to different treatments with different concentrations of Ag^+^, chemical treatment with NaBH_4_, then by freeze-drying. The characterization of obtained composite beads clearly showed the presence of AgNPs in the structure of the alginate and no other phase corresponding to ZnO or ZnNPs was obtained. The results confirm the presence of a porous structure due to the crosslinking of the alginate by Zn^2+^ and also due to the freeze-drying treatment. At higher concentrations of AgNPs, it was found the presence of nanoparticles on the outer surface of Zn-ALG corresponding to AgNPs. The application of these composite beads in the reduction of dyes in a simple system showed excellent results compared to the materials conducted in the literature using only low concentrations of NaBH_4_. The reaction time was as follows 480 s, 750 s, 750 s, and 840 s for the OG, MO, CR, and MB dyes, respectively. In the binary system, the reduction of the dyes was more selective towards the MB dye, following the electrostatic attraction between the electronic layer coming from the NaBH_4_ and the MB dye. It was shown that the use of the composite having a higher percentage of AgNPs (Zn-ALG(Ag 3%)) leads to strong leaching of Ag species in the reaction medium. Thus, in terms of stability and performance, the composite Zn–AlG(Ag 0.5%) was selected as the best catalyst, from where it was tested in six reuses without losing its performance. The antibacterial application showed encouraging results, in which Zn–AlG(Ag 1%), Zn–AlG(Ag 2%), Zn–AlG(Ag 3%) composites were selected as the most effective materials.

## Figures and Tables

**Figure 1 polymers-14-04829-f001:**
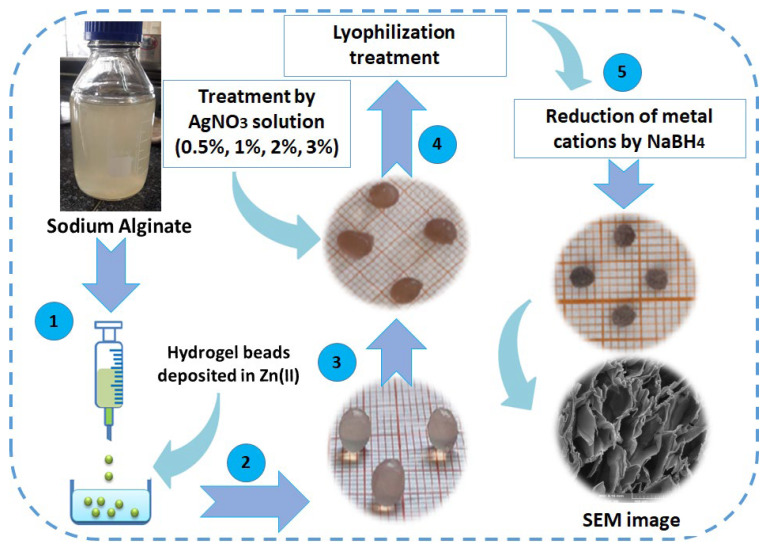
Preparation steps of the composite beads.

**Figure 2 polymers-14-04829-f002:**
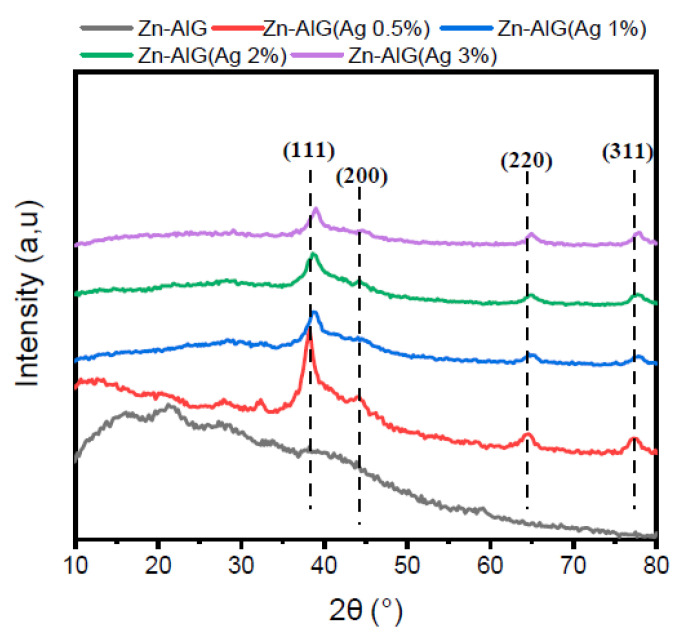
XRD patterns of obtained composite beads.

**Figure 3 polymers-14-04829-f003:**
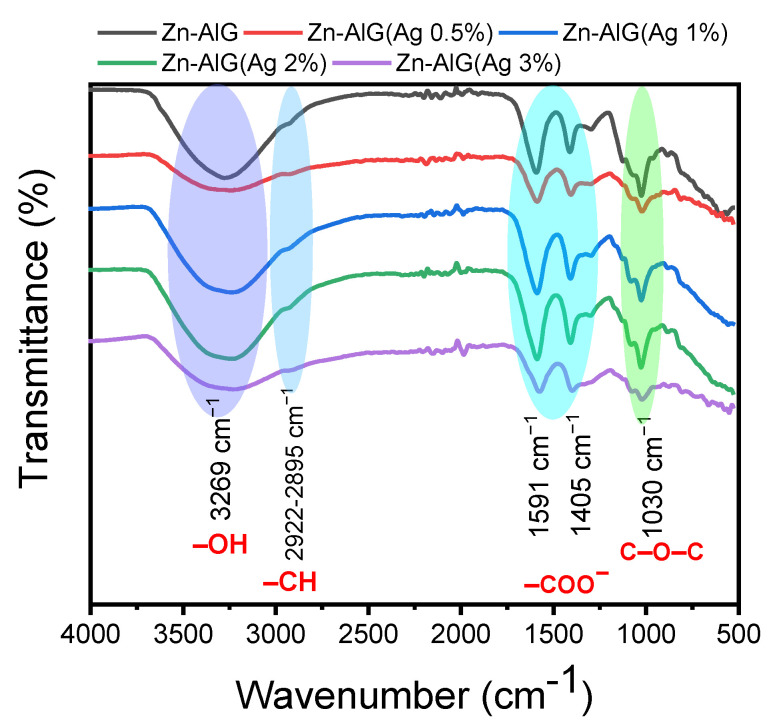
FTIR spectra of obtained composite beads.

**Figure 4 polymers-14-04829-f004:**
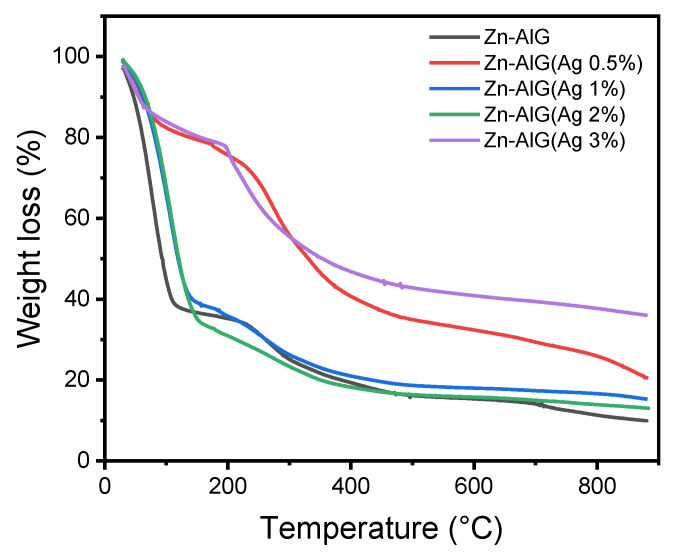
TGA curves of obtained samples.

**Figure 5 polymers-14-04829-f005:**
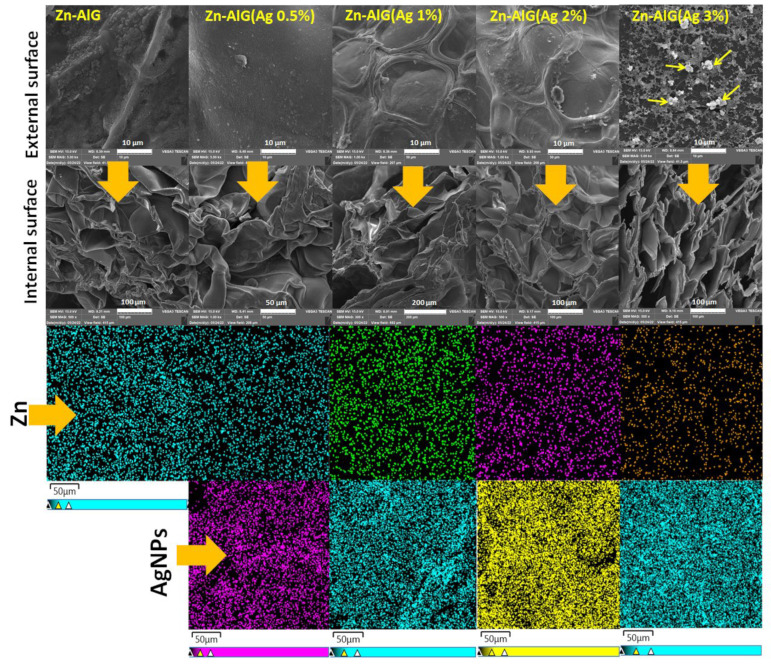
SEM/EDS mapping images of Zn–AlG (AgNPs) composite beads.

**Figure 6 polymers-14-04829-f006:**
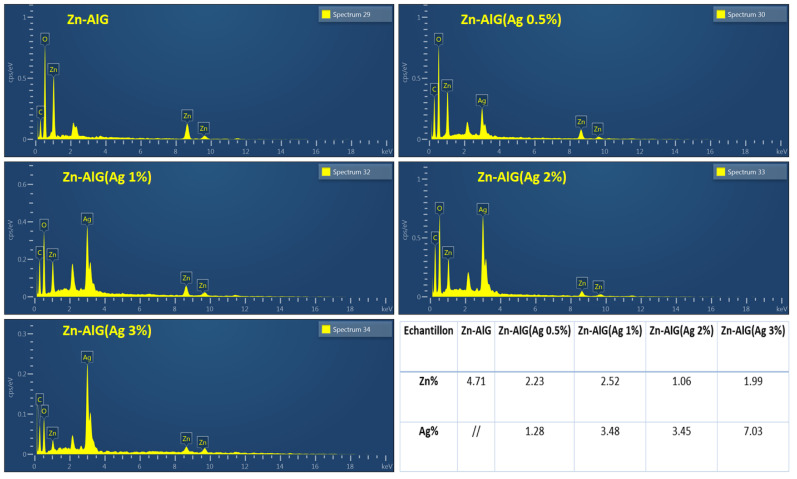
EDS of Zn–ALG(AgNPs) composite beads.

**Figure 7 polymers-14-04829-f007:**
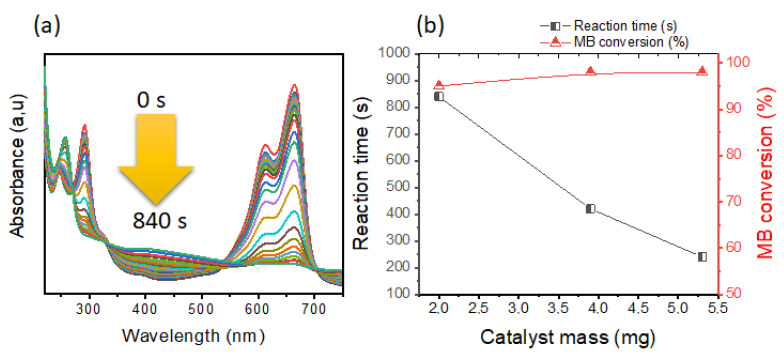
Mass effect of Zn–AlG(Ag 3%) catalyst, (**a**) UV-vis of BM dye during the reduction reaction, condition: one catalyst bead, [NaBH_4_] = 8 mM, [MB] = 0.1 mM. (**b**) Correlation between catalysts mass, reaction time and MB dye conversion.

**Figure 8 polymers-14-04829-f008:**
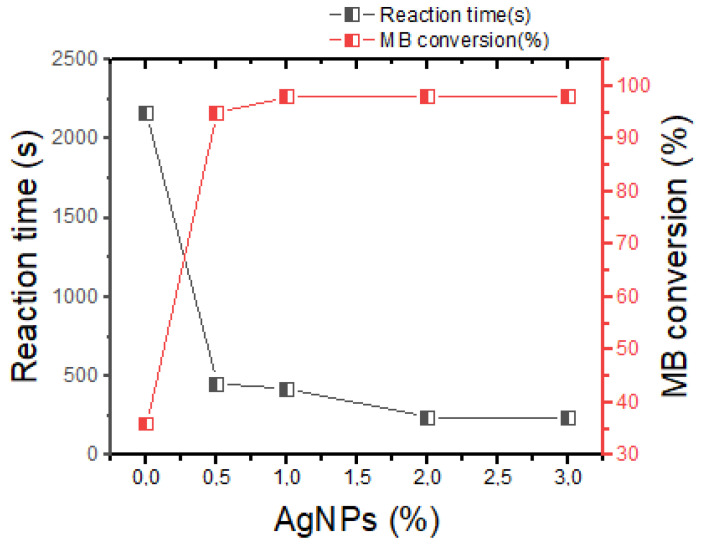
Correlation between AgNPs content, reaction time and MB conversion.

**Figure 9 polymers-14-04829-f009:**
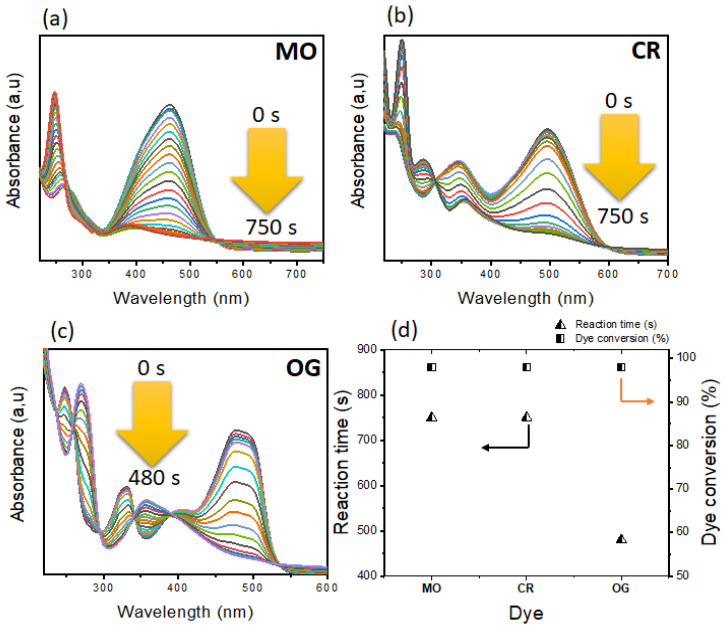
(**a**–**c**) UV-vis of different dyes (MO, CR and OG) during the reduction reaction. (**d**) Correlation between dye type, reaction time and dye conversion.

**Figure 10 polymers-14-04829-f010:**
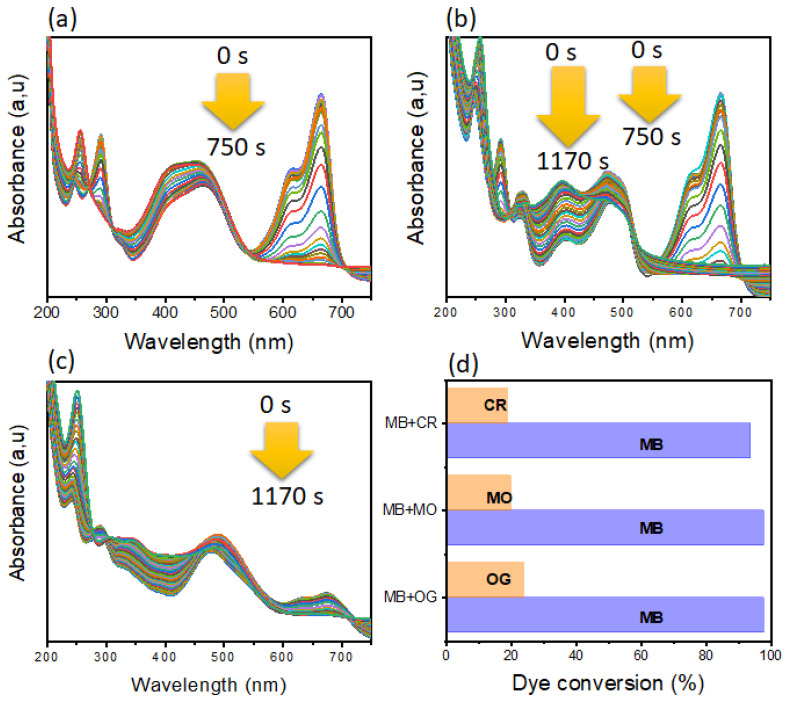
(**a**–**c**) UV-vis of dyes in a binary system, (**a**) MB/MO, (**b**) MB/OG and (**d**) MB/CR. (**d**) Conversion of different dyes in a binary system.

**Figure 11 polymers-14-04829-f011:**
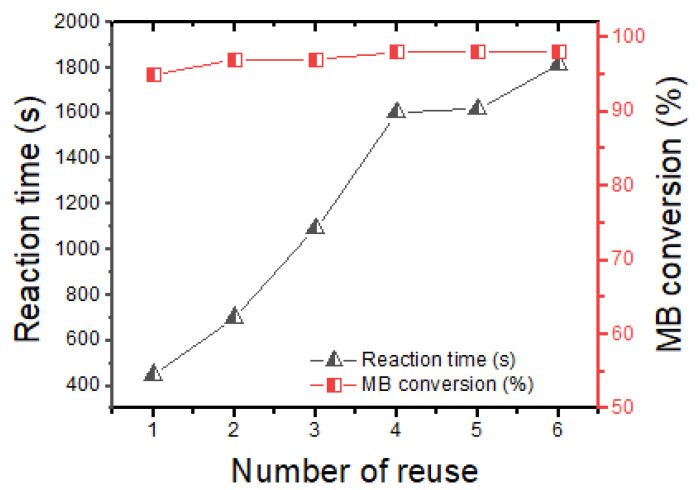
Reuse of the Zn-ALG(Ag 0.5%) catalyst.

**Figure 12 polymers-14-04829-f012:**
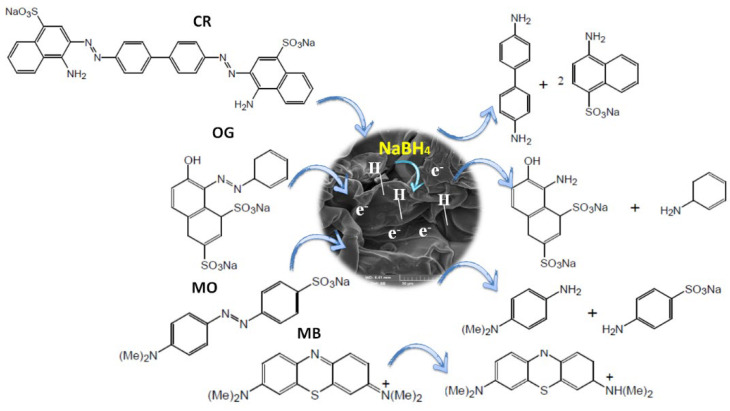
Proposed mechanism for catalytic reduction of different pollutants on the Zn-ALG(Ag 0.5%) catalyst.

**Table 1 polymers-14-04829-t001:** Antibacterial results.

	Zn–AlG (Ag 0.5%)	Zn–AlG (Ag 1%)	Zn–AlG (Ag 2%)	Zn–AlG (Ag 3%)	Zn–AlG	Na–AlG	Gent
*E.c*	15 mm	19 mm	17 mm	25 mm	20 mm	19 mm	26 mm
*P.a*	20 mm	16 mm	22 mm	19 mm	0 mm	19 mm	27 mm
*S.a*	17 mm	18 mm	16 mm	18 mm	4 mm	19 mm	28 mm

## Data Availability

Not applicable.

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
