# Peer review of "Catalytic Reduction of Dyes and Antibacterial Activity of AgNPs@Zn@Alginate Composite Aerogel Beads"

_polymers, 2022, doi:10.3390/polym14224829_

Round 1

Reviewer 1 Report

The paper presents an interesting experiental work on the preparation and characterization of hydrogel beads based on alginate crosslinked with Zn and filled with silver nanoparticles for the reduction of dyes. This is an interesting topic and the research is important in this field. I recommend minor revisions

1)     Please underline the novelty of the work in the Introduction

2)     Figure 6: please add the EDS map to see the eventual homogenous distribution of Zn ions.

3)     The conclusions are too long; please divide them by points

4)     The manuscript does not cite many papers published in Polymers. This may give an impression to our readers that the work is outside the scope of the journal. Therefore, I recommend to rework the list of references and cite papers published in Polymers or in other related MDPI journals (such as Nanomaterials and Materials).

Author Response

Reviewer 1

Comments and Suggestions for Authors

The paper presents an interesting experimental work on the preparation and characterization of hydrogel beads based on alginate crosslinked with Zn and filled with silver nanoparticles for the reduction of dyes. This is an interesting topic and the research is important in this field. I recommend minor revisions

  • Please underline the novelty of the work in the Introduction

Reply

The introduction has been modified according to the reviewer's suggestions, please see the revised manuscript.

  • Figure 6: please add the EDS map to see the eventual homogenous distribution of Zn ions.

Reply

We have added EDS mapping as well as the discussions regarding this analysis. Please see the revised manuscript.

  • The conclusions are too long; please divide them by points.

Reply

The conclusion has been modified and lightened according to the reviewer's suggestions. Please see the revised manuscript

  • The manuscript does not cite many papers published in Polymers. This may give an impression to our readers that the work is outside the scope of the journal. Therefore, I recommend to rework the list of references and cite papers published in Polymers or in other related MDPI journals (such as Nanomaterials and Materials).

Reply

We have added some references from the Polymers MDPI journal to improve our manuscript. Please see the revised manuscript.

Reviewer 2 Report

1、  Please mark the main peaks on the FTIR spectrum.

2、  “It is clear that these bands were slightly shifted during the increase in the content of AgNPs, probably due to several interactions between the silver nanoparticles and the functional groups of the alginate.” was mentioned in the text (3.1.2.). Please give a detailed description of “several interactions”.

3、  Why Zn-AlG (Ag 3%) catalyst was selected to reduce MB, why not using other doping ration?

4、  “However, it was found that strong leaching of AgNPs takes place when using catalysts with higher content of AgNPs.”. How to prove this point?

5、  Please give the discussion of degradation mechanism. There are too many short paragraphs in this article. Can these contents be merged into one or two paragraphs?

6、  The TGA curve of Zn-ALG(Ag 0.5%) should be revised. The weight loss of this sample did not reach the equilibrium state.

7、  The language in both the title and context needs a revision.

8、   The references list should be checked. The following related refs. (or part of them) are suggested to be cited: https://doi.org/10.1016/j.colsurfb.2022.112443; https://doi.org/10.3390/biom12081110; https://doi.org/10.1007/s42114-022-00478-3.

Author Response

Reviewer 2

Comments and Suggestions for Authors

  1. Please mark the main peaks on the FTIR spectrum.

Reply

We have modified this figure, please see the revised manuscript.

  1. “It is clear that these bands were slightly shifted during the increase in the content of AgNPs, probably due to several interactions between the silver nanoparticles and the functional groups of the alginate.” was mentioned in the text (3.1.2.). Please give a detailed description of “several interactions”.

Reply

We have improved the discussions of this part, please see the revised manuscript.

  1. Why Zn-AlG(Ag 3%) catalyst was selected to reduce MB, why not using other doping ration?

Reply

In terms of conversion and reaction time, the Zn-AlG(Ag 3%) sample was selected as the best catalyst. But we noticed a coloring of the solution due to the strong leaching of AgNPs in the reaction medium.

  1. “However, it was found that strong leaching of AgNPs takes place when using catalysts with higher content of AgNPs.”. How to prove this point?

Reply

The leaching of AgNPs was confirmed by UV-Vis spectroscopy from which it was observed the appearance of a new band at 420 nm during the reduction reaction which confirms the presence of AgNPs in the reaction medium (see figure 1 in the attached file).

  1. Please give the discussion of degradation mechanism. There are too many short paragraphs in this article. Can these contents be merged into one or two paragraphs?

Reply

In the revised version of the manuscript, we have added a well-detailed degradation mechanism associated with a mechanism figure. Please see the revised manuscript.

  1. The TGA curve of Zn-ALG(Ag 0.5%) should be revised. The weight loss of this sample did not reach the equilibrium state.

Reply

Thank you for this valuable remark. It must be taken into account that above 900 °C the residue begins to degrade in the gas phase. In general, the curves of alginate-based composites are analyzed between 30-700°C to determine the different solid forms of the composites after their degradation.

  1. The language in both the title and context needs a revision.

Reply

The manuscript has been corrected, and the style has been improved, please see the revised version.

  1. The references list should be checked. The following related refs. (or part of them) are suggested to be cited: https://doi.org/10.1016/j.colsurfb.2022.112443; https://doi.org/10.3390/biom12081110; https://doi.org/10.1007/s42114-022-00478-3.

Reply

These references have been added to the manuscript please see the revised manuscript.

Round 2

Reviewer 2 Report

6. Authors are still recommend to revise the TGA figure. Although t above 900 °C the residue begins to degrade in the gas phase, why the curve (Zn-Alg, Ag 0.5%) different with others?

8. The references list should be checked. The following related refs. (or part of them) are suggested to be cited: https://doi.org/10.1016/j.colsurfb.2022.112443; https://doi.org/10.3390/biom12081110; https://doi.org/10.1007/s42114-022-00478-3.